# Imaging antiferromagnetic antiphase domain boundaries using magnetic Bragg diffraction phase contrast

Min Gyu Kim[1], Hu Miao[2], Bin Gao [1], S.-W. Cheong[1], C. Mazzoli[3], A. Barbour[3], Wen Hu [3], S.B. Wilkins[3], I.K. Robinson[2], M.P.M. Dean [2] & V. Kiryukhin[1]

Manipulating magnetic domains is essential for many technological applications. Recent breakthroughs in Antiferromagnetic Spintronics brought up novel concepts for electronic device development. Imaging antiferromagnetic domains is of key importance to this field. Unfortunately, some of the basic domain types, such as antiphase domains, cannot be imaged by conventional techniques. Herein, we present a new domain projection imaging technique based on the localization of domain boundaries by resonant magnetic diffraction of coherent X rays. Contrast arises from reduction of the scattered intensity at the domain boundaries due to destructive interference effects. We demonstrate this approach by imaging antiphase domains in a collinear antiferromagnet $Fe_2Mo_3O_8$, and observe evidence of domain wall interaction with a structural defect. This technique does not involve any numerical algorithms. It is fast, sensitive, produces large-scale images in a single-exposure measurement, and is applicable to a variety of magnetic domain types.

[1] Department of Physics and Astronomy, Rutgers University, Piscataway, NJ 08854, USA. [2] Condensed Matter Physics and Materials Science Department, Brookhaven National Laboratory, Upton, NY 11973, USA. [3] National Synchrotron Light Source II, Brookhaven National Laboratory, Upton, NY 11973, USA. Correspondence and requests for materials should be addressed to V.K. (email: vkir@physics.rutgers.edu)

Advances in information technologies have always been deeply rooted in the development of the hardware base. It is recognized that breakthroughs in materials research and discoveries of new device operation principles are necessary to sustain the current improvement rate in the miniaturization and operation speed of electronic devices in the coming decades[1]. Spintronics is a research direction that unifies several promising ideas for future devices based on utilization of the spin degrees of freedom[2]. Some of the spintronics concepts are already used in commercial devices, such as magnetic reading heads and magnetic memory cells[3]. Historically, spintronic devices were based on ferromagnets (FM), because the spins in these materials could be easily manipulated by an applied magnetic field[4,5]. In contrast, no convenient tools were readily available to control the properties of antiferromagnets (AFM) due to their zero net magnetization.

Recent experimental and theoretical breakthroughs have radically changed this situation. Convenient methods for all-electrical AFM domain switching (e.g., utilizing the Néel spin-orbit torque) were discovered[6], and devices containing AFM as key parts were demonstrated[4,5]. It was realized that AFM systems present several important advantages over the FM counterparts. They typically operate at THz frequencies, which is up to two orders of magnitude faster than typical FM[4,5,7]. AFM devices do not produce significant stray fields, and are largely immune to external magnetic noise, making them ideally suitable for device miniaturization. In fact, several recent reviews on what is now called Antiferromagnetic Spintronics argue that AFM materials could form the future of spintronics[4,5].

The ability to image magnetic domains in AFM materials is of key importance for Antiferromagnetic Spintronics. While imaging techniques for the domains on the surfaces of FM materials have been available for more than a century, AFM domain imaging is a relatively new development. There is a multitude of possible AFM spin arrangements, and the corresponding AFM domain types. Imaging techniques based on the polarization dependence of the observed signal can image domains distinguished by the direction of the spin alignment, or by the helicity of the spin rotation. Examples include X-ray photoemission electron microscopy[8], and scanning magnetic X-ray diffraction[9,10]. Such techniques are not applicable to the simplest kind of the AFM domains that may occur in any antiferromagnet, the so-called antiphase domains. Two such domains differ by the reversal of the direction for all the spins, and, therefore, by the

sign of the appropriate AFM order parameter. For example, the up-down pattern becomes down-up on crossing the antiphase domain boundary. In addition, the described techniques do not work for collinear spin arrangements. Conceptually, antiphase domains and collinear AFM are the most basic building blocks available for Antiferromagnetic Spintronics. One or both of these two cases applies to a vast number of known AFM.

Our new imaging technique utilizes detection and localization of the domain boundaries for identification of the domain pattern. It is applicable to a large variety of AFM domain types, including the antiphase domains, suitable for collinear and noncollinear systems, and unrestricted by any symmetry requirements. We demonstrate this technique by imaging the antiphase AFM domain boundaries in a collinear antiferromagnet $Fe_2Mo_3O_8$. In our experiments, 5 μm spatial resolutions are achieved in a system with 0.9 $\mu_B$ effective net ordered magnetic moment ($\mu_B$ is Bohr magneton), and images of $0.3 \times 0.3$ mm area are obtained in 1 s. We argue that the spatial resolution of this technique can be improved to submicron scales, the sensitivity can be pushed into the 0.1 $\mu_B$ range, and $10^{-2}$ s time resolutions are achievable. No algorithmic image reconstruction of any kind is needed.

## Results

**Experimental setup.** The main idea of our approach is illustrated in Fig. 1. The magnetic Bragg peak is measured in specular reflection geometry. The coherent X-ray beam emits from a circular pinhole located close to the sample, the reflected intensity is measured by a remote area detector. Consider a small portion of the beam of size $d$ reflecting directly off a sharp AFM antiphase domain boundary, as shown in Fig. 1a. The phase of the scattered photons in a magnetic Bragg reflection changes by π on crossing the boundary due to the phase change of the AFM order parameter. The resulting destructive interference will reduce the Bragg signal coming from a certain vicinity of the boundary. Consequently, a fringe pattern with a dark line in the center will be produced at the detector. The details of the pattern are determined by the scattering geometry, the variation of the phase across $d$ in the incident beam, as well as by the beam divergence. In our experiment, the pinhole produces a circular Airy fringe pattern of bright and dark rings on the sample surface of the diameter $D \gg d$. We observe more than 50 bright rings, covering an approximately 300 μm radius area on the sample, see

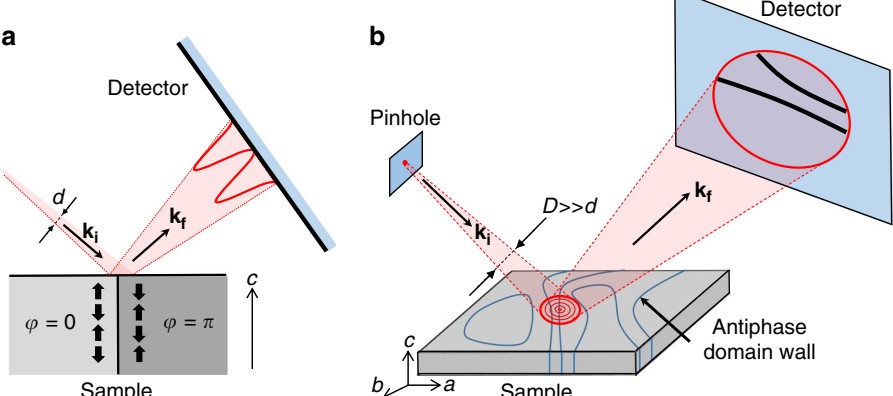

**Fig. 1** Schematic experimental setup. **a** Diffraction of a small beam off an antiphase magnetic boundary. Black solid arrows indicate the effective magnetic order probed in our experiment, $\varphi$ is the phase of the magnetic order parameter, $\mathbf{k_i}$ and $\mathbf{k_f}$ are the incident and outgoing wave vectors. A fringe pattern with a dark central line is produced on the area detector. **b** The full experimental setup, showing the large incident beam spot on the $ab$ surface of the sample, and the image of the domain boundaries (dark lines) obtained on a remote area detector. Beam diameters $D$ and $d$ are explained in the text

Supplementary Fig. 1. The incident beam phase changes by $\pi$ between the adjacent rings, setting the effective length scale $d$ for the destructive interference in the reflected beam. On the detector, we therefore expect to see a direct, magnified image of the antiphase domain boundaries as dark lines surrounded by a pattern of fringes, see Fig. 1b. Very broadly, there is an analogy to propagation-based phase contrast X-ray imaging in the edge enhancement regime[11–14]. However, the key components of our technique, such as the use of structured beam, the origin of the phase contrast (antiphase signals vs. refraction), experimental geometry (reflection vs. in-line imaging), as well as innate direct sensitivity to the antiferromagnetic signal are all distinctly different.

**Sample**. $Fe_2Mo_3O_8$ crystallizes in a hexagonal $P6_3mc$ structure consisting of honeycomb Fe layers separated by nonmagnetic Mo-O sheets, stacked along the $c$ axis[15]. Below $T_N \approx 60$ K, the $Fe^{2+}$ spins order and form a collinear AFM structure shown in Fig. 2a, which only supports antiphase domains[16]. $Fe_2Mo_3O_8$ and several

isostructural compounds have recently attracted significant attention due to the plethora of unusual effects originating from coupling of magnetism to the crystal lattice. They include giant magnetoelectricity[17,18], unconventional electromagnon excitations[19], giant thermal Hall effect[20], exotic axion-type magnetoelectric susceptibility[21], as well as nonreciprocal spectroscopic effects in the THz range[22]. AFM domain behavior is clearly relevant to all these intriguing properties.

**Experimental results**. The AFM structure of $Fe_2Mo_3O_8$ is probed directly by resonant magnetic X-ray diffraction at the (0,0,1) Bragg peak that has only magnetic and no lattice contribution. Figure 2a shows the resonant enhancement of this peak at the Fe $L_{III}$ absorption edge. The resonance occurs at $E = 708.3$ eV (wavelength $\lambda = 1.750$ nm), and the subsequent (0,0,1) peak measurements are performed at this energy. The corresponding fluorescence scan is shown in Fig. 2b. The temperature dependence of the (0,0,1) peak intensity, which measures the square of the AFM order parameter, is shown in Fig. 2c. As expected, no signal is observed above $T_N$. $Fe_2Mo_3O_8$ has two inequivalent Fe sites with different magnetic moments, 4.8 $\mu_B$ and 4.2 $\mu_B$[23]. As a result, each Fe layer has a small net magnetization. These ferrimagnetic layers are stacked antiferromagnetically along the $c$ axis. The (0,0,1) peak is sensitive only to the $c$-axis variation of the magnetic moment integrated in the $ab$ plane. While the peak's intensity has an additional magnetic contribution due to the small buckling of the Fe planes, it is convenient to think about the (0,0,1) peak as a direct measure of the AFM ordering of the Fe planes' net moments along the $c$ axis. This effective one-dimensional AFM order is shown by purple arrows in Fig. 2a. The domain boundaries probed in this work lie in the $ab$ plane, separating the up-down and down-up antiphase domains as shown in Fig. 1a. As explained in Supplementary Note 1, the domain boundary width is expected to be nm-scale because of the large magnetic anisotropy.

A typical pattern observed on our detector at the (0,0,1) magnetic peak position at $T = 30$ K is shown in Fig. 3a. Wavy dark lines surrounded by a fringe pattern are clearly seen. (See Fig. 4b inset for the detailed image of the fringe pattern.) The Bragg condition at (0,0,1) must hold in order to observe the image everywhere on the detector. This is ensured by the small X-ray penetration depth of $\mu = 0.1$ μm at the $L_{III}$ absorption edge, and the corresponding Bragg peak broadening of the order of $\lambda/\mu \sim 1°$. A slight tilt of the sample extends the Bragg condition validity region either towards larger or smaller scattering angles (top and bottom detector parts, respectively). Stitching together a few (three to five) such images produces a larger visible field of view shown in Fig. 3b. To establish the magnetic features in the observed signal, we have studied the same surface region using the structural (0,0,2) Bragg peak with X-ray of double the energy. This off-resonance setup preserves the same scattering geometry, and the signal has no magnetic contribution. Figure 3c shows the single-exposure (no scanning of any kind) image undertaken these conditions. The usable image area is significantly reduced in the vertical detector direction, corresponding to the reduced range of the scattering angles for which the Bragg condition holds. This is consistent with the increased penetration depth (0.65 μm) of the 1416.6 eV X-rays used in this measurement. To get the usable image over the full detector area, the sample tilting and image stitching procedure described above was used, the result is shown in Fig. 3d. No wavy lines are observed, but several small ring-like structures are clearly seen. They are of structural origin, and persist above $T_N$. These structural features are also observed in Fig. 3a, b, and can be discarded, leaving the wavy lines, which are therefore of a purely magnetic nature. As shown

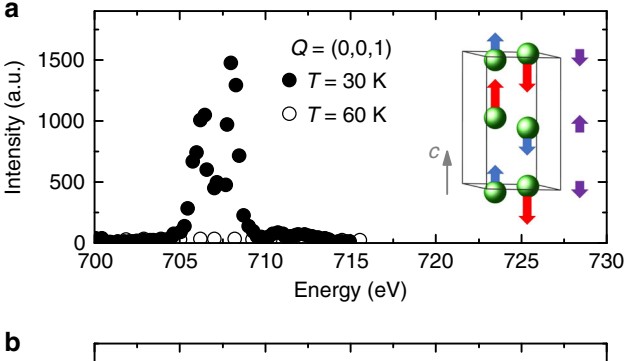

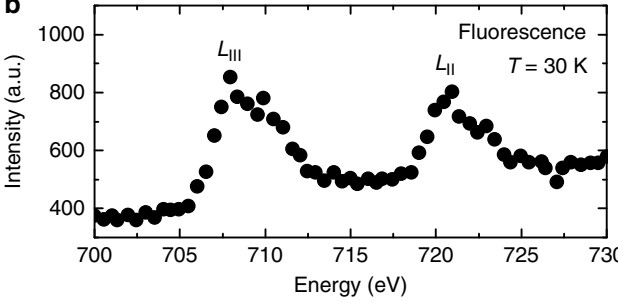

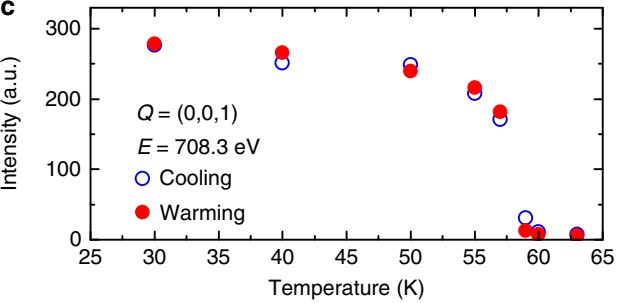

**Fig. 2** Resonant magnetic X-ray diffraction in $Fe_2Mo_3O_8$. **a** Right. The magnetic order in $Fe_2Mo_3O_8$. Blue and red arrows represent $Fe^{2+}$ magnetic moments at two different crystallographic sites, the difference between the moment magnitudes is exaggerated. Purple arrows represent the net magnetic moments of the Fe planes stacked along the $c$ axis. Left. Magnetic resonance at the (0,0,1) magnetic Bragg peak. **b** X-ray fluorescence through the $L_{II}$ and $L_{III}$ absorption edges. **c** Temperature dependence of the magnetic (0,0,1) peak intensity, representative of the magnetic order parameter behavior

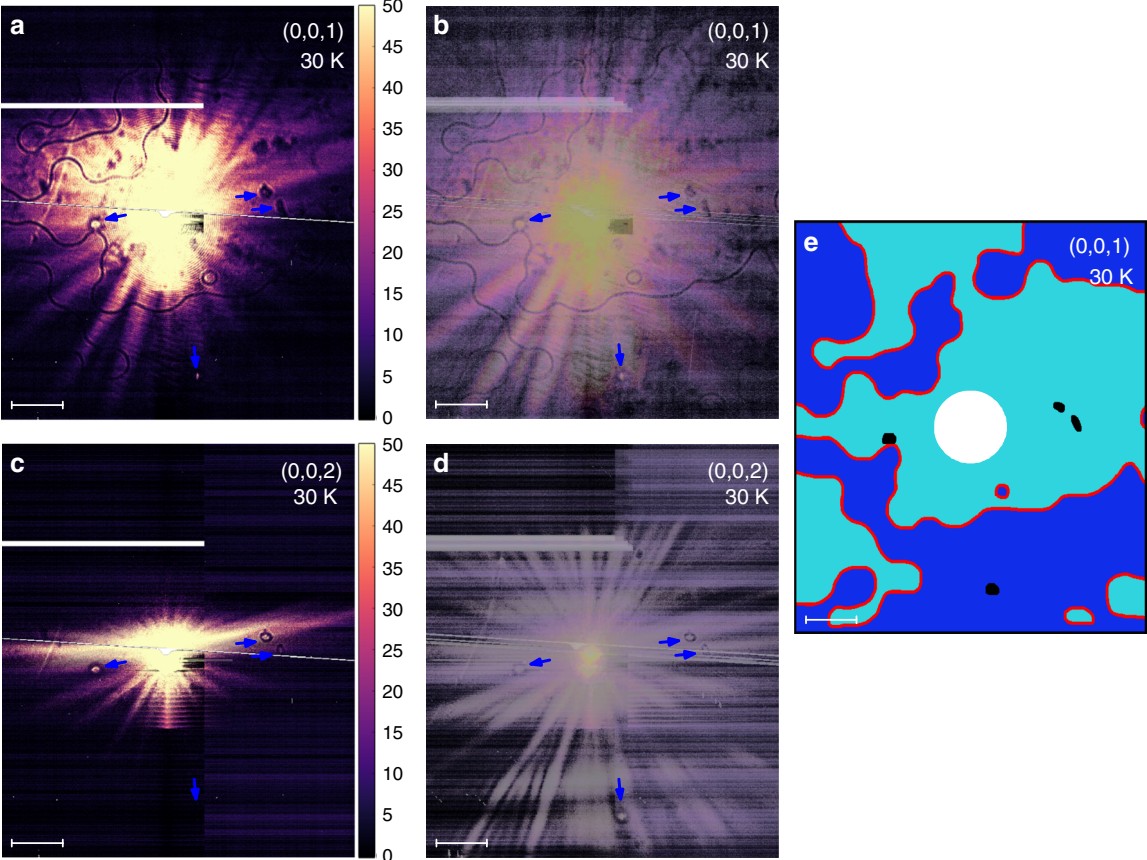

**Fig. 3** Images of AFM antiphase domain boundaries. Single-exposure (**a**), (**c**), and stitched-together images (**b**), (**d**) at the magnetic (0,0,1) and structural (0,0,2) Bragg peak positions, as indicated. Black wavy lines in **a** and **b** are images of the antiphase domain boundaries. Arrows show structural defects. Horizontal scale bars (50 μm) refer to the distances on the sample surface, as explained in the text. The reconstructed AFM domain pattern is shown in **e**. Black ovals indicate structural defects, white circle covers the area inaccessible due to detector saturation. All the detector images in this paper are elongated vertically by the factor of 1.15 to compensate for the beam footprint size effect and produce uniform magnification, see the Methods section. Vertical color scale-bar units are arbitrary

below, these lines represent real features on the physical surface of the sample. We therefore conclude that the lines image the antiphase magnetic domain boundaries. The maximum divergence angles in our reflection geometry are small (limited by ∼ 2°), and therefore the observed detector patterns are direct, undistorted (up to a footprint correction), magnified images of the domain boundaries on the sample surface. Using the positions of these boundaries, one can easily construct the corresponding domain patterns, as shown in Fig. 3e. The observed large typical domain size of the order of 100 μm indicates high structural quality of the investigated samples.

Larger-area images can be obtained by stitching together overlapping images obtained at several different positions on the sample surface. An example of such an image and the reconstructed domain pattern are shown in Supplementary Fig. 2. The success of the stitching procedure following small sample displacements validates the projection imaging procedure and gives an estimate of the magnification. While taking these images, an interesting effect was observed. A prolonged X-ray exposure is found to modify the diffraction properties of a small, 2–3 μm-diameter area in the center of the beam. This area is revealed in images taken using a different beam position on the sample as a bright spot, i.e. it exhibits a higher X-ray reflectivity than the virgin surface. We have painted a grid of such spots, separated by 10 μm, see Fig. 4. Spots removed by more than 100 μm from the center of the beam are clearly seen, confirming that a large sample area is imaged simultaneously. This effect provides a way of

obtaining the length scale on the sample surface, and the image magnification, $m = 66$. The scale bars shown in all figures refer to the distances on the sample surface, calculated using this magnification. Importantly, the dark lines are pinned to the coordinate grid formed by the spots on the sample surface, and therefore represent real physical features on the sample surface. The spots persist at low temperatures, and are unaffected by heating above $T_N$. However, the virgin surface is restored and the spots disappear on brief annealing at 170 K. The origin of this effect is unclear. Interesting scenarios, such as a photoinduced phase transition[24], or mundane explanations, such as ice accumulation, can all be considered at this stage. Further investigation of this effect will be subject of future studies.

The image magnification is largely set by the ratio of the sample-detector and sample-pinhole distances, and moderately increased by the beam divergence effects. The total observed width of the dark wavy lines (together with any surrounding fringes) produced by the domain boundaries corresponds to about 5 μm on the sample surface, setting the experimental spatial resolution and the minimum size for the observable domains. (The actual boundaries are much thinner than 5 μm.) Calculation of the actual fringe patterns for the highly structured, divergent incident beam goes beyond the scope of our work. Experimentally, the effective size of the region producing the central dark line is 2 μm. It matches the distance between the Airy diffraction rings in the incident beam on the sample surface, and justifies the arguments on what sets the length scale $d$ for the destructive

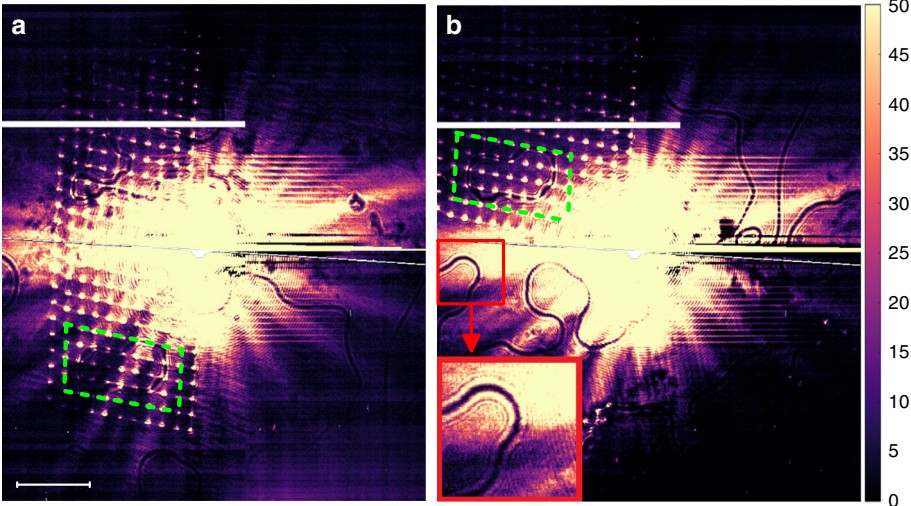

**Fig. 4** Coordinate grid on the sample surface. **a** A single-exposure image showing an array of bright spots that were produced by a prolonged beam exposure at numerous positions on the sample surface. The vertical and horizontal distance between the adjacent dots is 10 μm. In this measurement, the whole cryostat was moved to achieve different beam positions on the sample for spot burning. This gives rise to the tilt of the axes in the spot array. The beam can also be moved on the sample surface by the pinhole translation with little axis tilt, but a smaller motion range is achievable in this case. **b** Image was taken at a different position of the beam on the sample surface. Dashed green lines forming a parallelogram in each figure go through the same spots on the sample surface in both images, and the two parallelograms cover the same physical area on the sample surface. The dark lines inside the parallelograms go through the same physical coordinates in both **a** and **b**. This provides a definite evidence that the lines in the images result from the corresponding features on the physical surface of the sample. A blowup of a small image region demonstrating the fringe pattern surrounding the central dark line is shown in the lower left corner in **b**. Regularly-spaced horizontal straight lines are artifacts due to detector saturation. Horizontal scale bars (50 μm) refer to the distances on the sample surface. Vertical color scale-bar units are arbitrary

interference from the domain boundary in Fig. 1a. The distance between the diffraction rings can be reduced by decreasing the sample-pinhole distance, increasing the pinhole size, or increasing the X-ray energy (not applicable for resonant diffraction). A factor of five reductions should be achievable by these means, leading to sub-μm resolutions. Higher resolutions are expected for larger-energy absorption edges. Advanced X-ray optical elements, such as zone plates, may provide further opportunities for improving resolution, as may be necessary for advanced spintronics applications. Larger-area single-exposure images are also achievable at reduced resolution for larger sample-pinhole distances, making this technique highly configurable. Studies of these opportunities are highly desirable.

An important advantage of this technique is acquisition of a large-area domain pattern in a single-exposure measurement. The (0,0,1) images shown here were collected for 2 s, but usable data could be measured in 0.5 s. The structure factor of the (0,0,1) peak in $Fe_2Mo_3O_8$ corresponds to the 0.9 $\mu_B$ effective ordered moment per Fe atom. With minute-scale exposures, effective moments as small as 0.1 $\mu_B$ should be accessible for imaging. For a compound with fully ordered Fe moment, acquisition times in the $10^{-2}$ s range are feasible. Thus, our technique makes possible time-resolved, in-situ studies of magnetic domain patterns. These studies can be performed under various conditions, such as applied electric or magnetic fields, thermal and electric currents, etc. In our experiment, we have observed the temperature evolution of the magnetic domain patterns in real time, as the sample was warmed up and cooled down through the Néel transition. Some of the obtained images are shown in Supplementary Fig. 3. The structural defects in the image provide the reference frame, allowing exact identification of the investigated sample area. We find that the domain patterns don't change during both heating and cooling below $T_N$, only the whole pattern motion reflecting the cryostat's thermal contraction is observed. However, a completely new domain pattern forms on each

cooling through $T_N$. Figure 5 shows such four different domain patterns. Interestingly, in three cases out of four, magnetic domain boundary forms at the structural defect location shown with an arrow in this figure. This indicates that structural defects may serve as nucleation or pinning centers during the formation of antiphase AFM domain walls. We note that strong defect pinning is typical for narrow domain walls in FM, where such pinning plays a key role in the technological applications[25]. Further studies establishing the nature of the observed structural defects, their interaction with the antiphase domain walls, as well as higher experimental statistics are necessary to establish whether similarly strong effects occur in AFM.

## Discussion

In collinear AFM, antiphase domains have been studied only for several very special cases. One available technique is polarized neutron diffraction topography. In specific low-symmetry structures involving non-equivalent environments for the up and down spins, the structure factors for the mixed (nuclear plus magnetic) Bragg reflections may depend on the AFM order parameter sign[26]. Only a few successful antiphase domain measurements by this technique have been reported[27–29]. Spatial resolution of only about 0.1 mm can be achieved, and hours of exposure are required. Another technique is nonlinear optics second-harmonic generation (SHG) imaging[30]. It is based on the interference of the time-invariant (lattice) and time-noninvariant (magnetic) contributions, and also works only in low-symmetry systems. Inversion symmetry, in particular, is not allowed. Unambiguous interpretation of the observed SHG signal is often difficult to achieve, the signals are typically low, and theoretical analysis is complicated. As in the neutron diffraction case, only a few successful measurements of antiphase AFM domains have been reported by SHG[30]. The obtained spatial resolutions are of the order of 20 μm, and typical exposures take minutes. It is clear

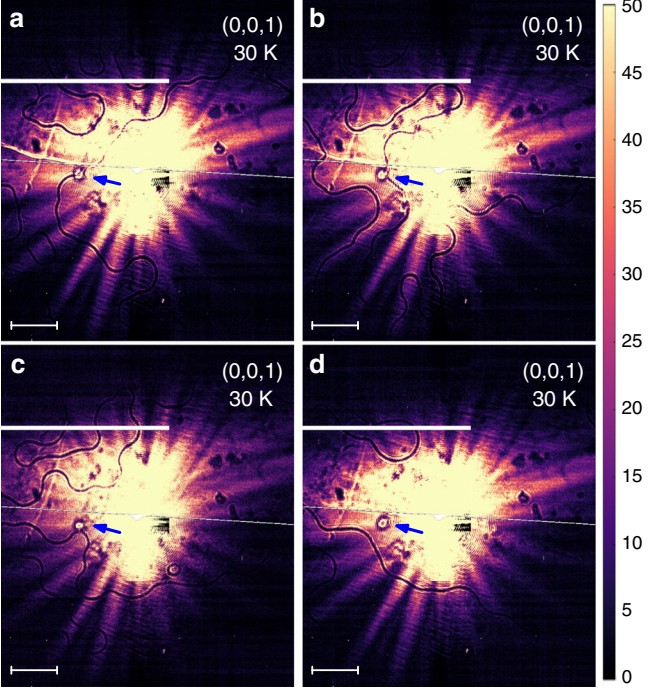

**Fig. 5** AFM domain boundaries after several different coolings from above the Néel temperature. The same sample area is shown in panels **a–d**. Arrows indicate the position of a stationary structural defect, used as a reference. Horizontal scale bars (50 μm) refer to the distances on the sample surface. Vertical color scale-bar units are arbitrary

that a very favorable combination of low magnetic and structural symmetry is required for the both described techniques to work. Finally, antiphase domain walls have been observed on a nanoscale in several natural and synthetic magnets using scanning probe techniques[31,32]. These techniques are slow, don't allow single-exposure imaging, and are limited to either nanometer-scale total image area, or to specific systems made of ferromagnetic layers.

There is a wide range of coherent X-ray imaging techniques[11–14]. Existing methods utilizing reflection geometry[14] require complex image reconstruction procedures. It is unknown whether their application for AFM domain imaging is realistic or, indeed, possible. Many transmission techniques use phase contrast arising from interference of X-rays scattering from adjacent regions with different properties. Examples include transport of intensity methods utilizing both extended and small sources[12,13]. Transmission magnetic phase contrast imaging of ferromagnetic domains has been achieved by Fourier transform holography[33], as well as by computationally demanding resonant X-ray ptychography[34]. None of these techniques are applicable to AFM antiphase domains on the surface of bulk samples. In this work, we utilize a reflection geometry to achieve computation-free direct imaging of AFM antiphase domain walls in thick samples, and probe the AFM order parameter directly by use of resonance at an antiferromagnetic Bragg peak, making our approach distinctly different.

The domain wall imaging technique described here can be applied to various types of magnetic domain boundaries, not just to the antiphase domains, as long as there is some phase difference between the portions of the beam scattered off the adjacent domains. Structural domain boundaries of various types should also be suitable objects for this technique. The important requirement is that the appropriate Bragg reflection is obtainable from the sample surface of interest at the appropriate X-ray

energy. It is satisfied for the systems with relevant periodicity $p$ larger than $\lambda/2$. For the Fe $L_{III}$ edge, $p$ is limited by 0.88 nm. Larger ranges of $p$ are accessible at the higher-energy edges of heavier elements. In the initial technique application reported here, images of $0.3 \times 0.3$ mm area were obtained in a second with 5 μm resolution. Significantly better spatial (submicron) and time ($10^{-2}$ s) resolutions can be achieved by rather straightforward adjustments described above. Importantly, in-situ measurements under various conditions, such as applied electric or magnetic fields, and thermal and electric currents can be carried out, which is relevant to potential technological applications. We believe that this technique opens new avenues for investigation of the fundamental properties of magnetic systems, structural and magnetic transitions, as well as for applied research.

## Methods

**Sample synthesis.** $Fe_2Mo_3O_8$ single crystals were synthesized using a chemical vapor transport method, as described in ref.[17]. The samples are hexagonal plates, with typical size of 1 mm. They exhibited natural mirror-like *ab* surfaces.

**X-ray diffraction.** Resonant magnetic X-ray diffraction measurements were carried out at the Coherent Soft X-ray Scattering (CSX) beamline, National Synchrotron Light Source II, Brookhaven National Laboratory. X-ray beam, polarized in the scattering plane, was collimated by a pinhole 5 μm in diameter located 6.5 mm before the sample, after which the beam is essentially coherent. Useful image is collected from the sample area on which the Airy diffraction pattern is produced by the pinhole, setting the requirements on the necessary beam coherence. The higher the beam quality (brilliance and coherence), the larger sample area could be imaged. Sample quality also affects the size of this area. In our measurements, the diameter of the area imaged in one exposure was between 200 and 300 μm. The signal was recorded by an in-vacuum CCD area detector (Berkeley Fast CCD, up to 100 Hz readout, $960 \times 960$ pixels, $30 \times 30$ μm pixel size, no polarization discrimination), located 34 cm away from the sample. The sample was mounted on a multi-circle in-vacuum diffractometer in a helium-flow cryostat. X rays in the Fe $L_{II}$ and $L_{III}$ energy range ($E = 700–730$ eV), as well as at $E = 1416.6$ eV were used. The CCD detector measures the intensity in energy-dependent instrument units. They are not calibrated to the X-ray photon count at the moment, and therefore are listed as arbitrary units in the figures. X-ray scattering was measured in a specular reflection geometry off the native *ab* surface of the sample, the scattering angle $2\theta$ was 121°. In this geometry, the direct image of the sample surface observed on the detector is compressed along the detector's vertical direction by the factor of $\sin(\theta)$ = 0.87 due to the beam footprint size effect. All the detector images shown in this paper are elongated vertically by the factor of 1.15 to compensate for this compression. Thus, the images feature identical length scale bars for the vertical and horizontal directions (uniform magnification). We were mainly interested in the weak scattering signal spread over the entire detector area. Therefore, measurements were done in the regime in which the detector area at the center of the Bragg peaks was saturated. To obtain the quantitative energy and temperature dependencies of the (0,0,1) peak intensity shown in Fig. 2a, c, the signal was measured slightly off the peak center, at the scattering vector (0,0,1- $\delta$), where $\delta = 5 \times 10^{-4}$.

## Data availability

The data that support the findings of this study are available from the corresponding author upon reasonable request.

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

## Acknowledgements

Work by the Rutgers group, including sample growth and X-ray scattering, was supported by the U.S. Department of Energy (DOE) under Grant No. DOE: DE-FG02-07ER46382. This research used resources at the 23- ID-1 beamline of the National Synchrotron Light Source II, a DOE Office of Science User Facility operated for the DOE Office of Science by Brookhaven National Laboratory under Contract No. DE-SC0012704. Work in the Condensed Matter Physics and Materials Science Division was supported by the DOE Office of Science, Office of Basic Energy Sciences, under Contract No. DE-SC00112704. M.P.M.D. acknowledges support by the DOE, Office of Basic Energy Sciences, Early Career Award Program under Award No. 1047478.

## Author contributions

V.K. conceived the idea and designed the study. B.G. and S.W.C. synthesized the samples. C.M., H.M., M.G.K., M.P.M.D., A.B., W.H., S.B.W. and V.K. carried out X-ray measurements. M.G.K., H.M., C.M., I.K.R., M.P.M.D. and V.K. analyzed the data. All authors discussed the results and contributed to the manuscript.

## Additional information

**Competing interests:** The authors declare no competing interests.

