## [Peer Review File · Nature Communications]

Reviewers' comments:

Reviewer #1 (Remarks to the Author):

This article presents a study of antiferromagnetic anti-phase boundaries in the collinear antiferromagnet (AFM), Fe₂Mo₃O₈. Through the use of resonantly scattered coherent soft X-rays, this paper explores a modality to image anti-phase boundaries for the AFM order via local cancellation of the scattering intensity in the region of the domain wall.

The authors has worked to clarify the manuscript and address my previous concerns. Nature Communications is a far better forum for the publication of these results, which are largely technical and focused on the imaging technique. I am supportive of the publication of this manuscript.

Reviewer #2 (Remarks to the Author):

The paper by M.G. Kim et al reports on a new imaging approach for antiphase domains in antiferromagnets, applicable also to collinear antiferromagnets. In essence, the approach is projection imaging in a divergent beam, where a magnified image of the sample is obtained in the far field. More precisely, the projection image formed by a magnetic Bragg peak is observed, allowing to obtain sensitivity to the antiferromagnetic arrangement of the magnetic moments. The authors demonstrate very convincingly that under coherent scattering conditions, this arrangement allows to detect the presence of antiphase domains due to the fact that beams diffracted from atoms close to the antiphase domain wall will acquire a phase shift of π . They thus show up as dark lines with reduced contrast in the projection image. In a variety of experiments (e.g. moving the sample by a defined amount, patching fields of view, changing the temperature through the Neel temperature, imaging at a structural diffraction peak in comparison) and theoretically considering the domain wall width, the authors demonstrate that this must be the underlying contrast mechanism for the images in their experiments on Fe₂Mo₃O₈ crystallites.

The imaging experiments and the model for the underlying contrast mechanism presented by the authors are very convincing. I don't doubt that imaging of antiphase domain boundaries has indeed been achieved in an approach that can cover comparatively large areas, can look at the surface-near layers of bulk samples and requires no iterative computation.

The authors achieve a spatial resolution of 5 micrometers, image acquisition times on the order of one second in a system with a net magnetic contrast of 0.9 Bohr magneton. They argue that sub-micrometer resolution, 1/100 sec temporal as well as a sensitivity to 0.1 Bohr magneton contrast should be possible with their approach. This claim is hard to check but seems feasible.

The authors discuss a variety of potentially competing techniques which they find to be either not sensitive to antiphase domains and collinear spin structures (e.g. PEEM, scanning XRD) or significantly less performant (neutron and SHG based approaches). I agree with this assessment. They also include a comparison to scanning probe techniques, where antiphase domains in collinear antiferromagnets have been imaged and argue that their method has distinctly different performance parameters, in particular the possibility for a large field of view.

In my view, the paper by Kim et al constitutes a very convincing demonstration of a beautifully simple technique to image antiphase domains in antiferromagnets. While the importance of antiphase domains for antiferromagnetic spintronics may be a bit exaggerated in the introduction, it is nevertheless true that the use of antiferromagnetic materials is of increasing relevance in a variety of application contexts. It must also be said that even if the increase in future spatial resolution anticipated by the authors can indeed be realized, the resulting lengthscale is still large compared to many of the intrinsic lengthscales encountered in the physics of spintronics, such as

the spin diffusion length in metals. On the other hand, obtaining an overview picture of antiphase domains even at 1 μm resolution will be very useful for a variety of studies and applications.

In my view, the paper reports important and convincing new results and is of interest for the readers of Nature Communications. In my opinion, the paper is publishable in the present form.

Reviewer #3 (Remarks to the Author):

Min Gyu Kim and coworkers have found that a very simple coherent x-ray magnetic diffraction provided an image of antiferromagnetic domain walls (antiphase boundaries). As the authors give a stress, this is really surprising. I think that this finding is worth publication in Nature Communications.

On the other hand, I would doubt the potential of this technique. From pages 9 to 10, the authors appeal the advantages of this technique. They say that the technique can be applied to various types of magnetic domain boundaries. I partly agree with them, but this technique can be applied to rather limited materials. The magnetic modulation in most of magnetic materials are too short to satisfy the Bragg condition under the resonant condition.

The authors also mention that significantly better spatial (sub-micron) can be achieved. However, they say just 'by rather straightforward adjustments' without giving any strategy.

A similar 'overselling' is also found in the top of page 9. In my opinion, newly added discussion on the possible pinning effect of a 'structural defect' would not be confirming. First, the authors do not show the nature of the structural defect, indicated by an arrow in Fig. 5. In particular, the effective size (length scale) of the defect should be essential for pinning a narrow antiphase boundary. Second, according to their observations, the antiphase boundaries were likely immobile. I hence do not completely understand the meaning of 'pinning'. Third, three out of four 'defects' are apart from any antiphase boundaries in Fig. 3. I do not think that the statistics are high enough for discussing the 'pinning' effect.

These two points should be revised before publication.

I also make several technical comments:

The polarization of the incident x-ray should be described in the method section.

The specification of CCD should be described in the method section. The pixel size and number may be important factors.

Judging from the color scales attached to Figs. 3a and 3c, it seems that the signal levels were quite low. I would be skeptical that the boundary positions can be clearly reproduced from such a low-count image. The scale bars would not be accurate.

Color scales should be attached also to Figs. 4 and 5.

Because the scattering angle was about 120 degrees, the magnification should be anisotropic in all the obtained CCD images. I hence recommend that the distorted direction should be explicitly mentioned. For example, two-dimensional length scale bars could be attached to Figs. 3, 4, and 5.

We thank the Reviewers for their thorough, positive, and constructive reviews. We find all the criticisms useful, and all the points raised valid. We have modified our paper accordingly. Below, we copy all the reviews, provide our response, and give point-by-point account of the changes made. The changes are also highlighted in the manuscript. We trust that in the revised paper, all the raised concerns are addressed, and express hope that the Editor and the Reviewers will agree with us. Again, we thank everyone for their thorough work that helped us improve the paper.

Below, the comments of the Reviewers are copied in their entirety (broken only by our responses), all the criticisms and requested changes are individually listed, our response to each of them is given, and the changes made in the manuscript are described. We note that only Reviewer #3 requested changes.

Reviewer #1 (Remarks to the Author):

This article presents a study of antiferromagnetic anti-phase boundaries in the collinear antiferromagnet (AFM), Fe₂Mo₃O₈. Through the use of resonantly scattered coherent soft X-rays, this paper explores a modality to image anti-phase boundaries for the AFM order via local cancellation of the scattering intensity in the region of the domain wall.

The authors has worked to clarify the manuscript and address my previous concerns. Nature Communications is a far better forum for the publication of these results, which are largely technical and focused on the imaging technique. I am supportive of the publication of this manuscript.

Our Response. We thank the Reviewer for the support for our manuscript's publication, as well as for the previous constructive comments.

Reviewer #2 (Remarks to the Author):

The paper by M.G. Kim et al reports on a new imaging approach for antiphase domains in antiferromagnets, applicable also to collinear antiferromagnets. In essence, the approach is projection imaging in a divergent beam, where a magnified image of the sample is obtained in the far field. More precisely, the projection image formed by a magnetic Bragg peak is observed, allowing to obtain sensitivity to the antiferromagnetic arrangement of the magnetic moments. The authors demonstrate very convincingly that under coherent scattering conditions, this arrangement allows to detect the presence of antiphase domains due to the fact that beams diffracted from atoms close to the antiphase domain wall will acquire a phase shift of π . They thus show up as dark lines with reduced contrast in the projection image. In a variety of experiments (e.g. moving the sample by a defined amount, patching fields of view, changing the temperature through the Neel temperature, imaging at a structural diffraction peak in comparison) and theoretically considering the domain wall width, the authors demonstrate that this must be the underlying contrast mechanism for the images in their experiments on Fe₂Mo₃O₈ crystallites.

The imaging experiments and the model for the underlying contrast mechanism presented by the authors are very convincing. I don't doubt that imaging of antiphase domain boundaries has indeed been achieved in an approach that can cover comparatively large areas, can look at the surface-near layers of bulk samples and requires no iterative computation.

The authors achieve a spatial resolution of 5 micrometers, image acquisition times on the order of one second in a system with a net magnetic contrast of 0.9 Bohr magneton. They argue that sub-micrometer resolution, 1/100 sec temporal as well as a sensitivity to 0.1 Bohr magneton contrast should be possible with their approach. This claim is hard to check but seems feasible.

The authors discuss a variety of potentially competing techniques which they find to be either not sensitive to antiphase domains and collinear spin structures (e.g. PEEM, scanning XRD) or significantly less performant (neutron and SHG based approaches). I agree with this assessment. They also include a comparison to scanning probe techniques, where antiphase domains in collinear antiferromagnets have been imaged and argue that their method has distinctly different performance parameters, in particular the possibility for a large field of view.

In my view, the paper by Kim et al constitutes a very convincing demonstration of a beautifully simple technique to image antiphase domains in antiferromagnets. While the importance of antiphase domains for antiferromagnetic spintronics may be a bit exaggerated in the introduction, it is nevertheless true that the use of antiferromagnetic materials is of increasing relevance in a variety of application contexts.

It must also be said that even if the increase in future spatial resolution anticipated by the authors can indeed be realized, the resulting lengthscale is still large compared to many of the intrinsic lengthscales encountered in the physics of spintronics, such as the spin diffusion length in metals. On the other hand, obtaining an overview picture of antiphase domains even at 1 μm resolution will be very useful for a variety of studies and applications.

Our Response. Our enthusiasm for antiferromagnetic domains in spintronics originates from reading two very recent review papers on the subject that we quote in our paper (Refs. 4 and 5). We are glad to see the Reviewer's concluding comment that "the use of antiferromagnetic materials is of increasing relevance in a variety of application contexts". We are also glad to see the Reviewer's conclusion on the spatial resolutions of our technique, "obtaining an overview picture of antiphase domains even at 1 μm resolution will be very useful for a variety of studies and applications". Overall, these conclusions are very positive, and we thank the Reviewer for making them.

Reviewer #2, Concluding paragraph

In my view, the paper reports important and convincing new results and is of interest for the readers of Nature Communications. In my opinion, the paper is publishable in the present form.

Our Response. We thank the Reviewer for the description of our results as "important and convincing new results", our new technique as "beautifully simple" (in previous paragraphs), and for the recommendation to publish the paper in the present form.

Reviewer #3 (Remarks to the Author):

Min Gyu Kim and coworkers have found that a very simple coherent x-ray magnetic diffraction provided an image of antiferromagnetic domain walls (antiphase boundaries). As the authors give a stress, this is really surprising. I think that this finding is worth publication in Nature Communications.

Our Response. We thank the Reviewer for the conclusion that our finding is worth publication in Nature Communications.

Reviewer #3. On the other hand, I would doubt the potential of this technique. From pages 9 to 10, the authors appeal the advantages of this technique. They say that the technique can be applied to various types of magnetic domain boundaries. I partly agree with them, but this technique can be applied to

rather limited materials. The magnetic modulation in most of magnetic materials are too short to satisfy the Bragg condition under the resonant condition.

Our Response. We agree that our technique, as generally characteristic to all soft x-ray diffraction techniques, has limitations on the periodicity (p) of the observable magnetic structures. We have added a precise limit on such periodicity to the paper, both in general terms ($p > \lambda/2$), and numerically for the Fe L3 edge utilized in our work. On page 6, bottom, we now emphasize the importance of this limitation: “The important requirement...”, and write that “It is satisfied for the systems with relevant periodicity p larger than $\lambda/2$. For the Fe L_{III} edge, p is limited by 0.88 nm. Larger ranges of p are accessible at the higher-energy edges of heavier elements.” Importantly, this is the concluding paragraph of the paper, and a very prominent place to spell out technique’s limitations.

Reviewer #3. The authors also mention that significantly better spatial (sub-micron) can be achieved. However, they say just 'by rather straightforward adjustments' without giving any strategy.

Our Response. The quoted statement about straightforward adjustments is made in the concluding paragraph (page 7, top). In fact, the appropriate strategy is given earlier in the paper, on page 5, paragraph 2. Specifically, we write that “The distance between the diffraction rings can be reduced by decreasing the sample-pinhole distance, increasing the pinhole size, or increasing the x-ray energy (not applicable for resonant diffraction). A factor of five reduction should be achievable by these means, leading to sub-um resolutions. Higher resolutions are expected for larger-energy absorption edges. Advanced x-ray optical elements, such as zone plates, may provide further opportunities for improving resolution”. (Note, this text is not highlighted because it is not new). The Reviewer has a valid point that the reference to the straightforward adjustments in the concluding paragraph is quite remote from their description earlier in the paper. Fortunately, we believe this is easily fixable by adding “described above” to the sentence mentioning the adjustments in the concluding paragraph (page 7 top), so that the reader could return to the earlier text, if necessary.

Reviewer #3. A similar 'overselling' is also found in the top of page 9. In my opinion, newly added discussion on the possible pinning effect of a 'structural defect' would not be confirming. First, the authors do not show the nature of the structural defect, indicated by an arrow in Fig. 5. In particular, the effective size (length scale) of the defect should be essential for pinning a narrow antiphase boundary. Second, according to their observations, the antiphase boundaries were likely immobile. I hence do not completely understand the meaning of 'pinning'. Third, three out of four 'defects' are apart from any antiphase boundaries in Fig. 3. I do not think that the statistics are high enough for discussing the 'pinning' effect. These two points should be revised before publication.

Our Response. These are all good and valid points. In the revised version, we use careful statements about what we observed and what these observations evidence. We explicitly acknowledge that the

nature of the discussed structural defect is yet to be established, that our observations provide only the initial evidence of an interaction between the structural defects and antiphase AFM domain boundaries, that what they suggest is that structural defects may serve as nucleation or pinning centers during the domain formation, and that higher experimental statistics is necessary to establish the importance of the wall-defect interactions. We believe that these are all careful statements justified by our data. We would also like to note that these observations are of a supplementary character to the paper's main message (method for observation of antiphase AFM domain boundaries).

Specifically, the following text has been added (page 5, bottom): "Interestingly, in three cases out of four, magnetic domain boundary forms at the structural defect location shown with an arrow in this figure. This indicates that structural defects may serve as nucleation or pinning centers during the formation of antiphase AFM domain walls. We note that strong defect pinning is typical for narrow domain walls in ferromagnets, where such pinning plays a key role in the technological applications.[25] Further studies establishing the nature of the observed structural defects, their interaction with the antiphase domain walls, as well as higher experimental statistics are necessary to establish whether similarly strong effects occur in antiferromagnets."

Importantly, any strong statements about domain wall pinning have been removed. In particular, the revised abstract now mentions "evidence of domain wall interaction with a structural defect", which we believe is a careful and accurate statement.

Reviewer 3 has also made several technical comments. We thank the Reviewer for making them. The issues raised are fixed in the revised paper as follows:

Reviewer #3. I also make several technical comments:

The polarization of the incident x-ray should be described in the method section.

Our Response. Done. Page 7, "polarized in the scattering plane".

Reviewer #3. The specification of CCD should be described in the method section. The pixel size and number may be important factors.

Our Response. Done. Page 7, "Berkeley Fast CCD, up to 100 Hz readout, 960x960 pixels, 30x30 μm pixel size, no polarization discrimination".

Reviewer #3. Judging from the color scales attached to Figs. 3a and 3c, it seems that the signal levels were quite low. I would be skeptical that the boundary positions can be clearly reproduced from such a low-count image. The scale bars would not be accurate.

Our Response. The color scale bars are in arbitrary instrument units of our CCD detector. This is now explicitly mentioned in the revised figure captions. In addition, we added the following statement on the detector units in the Methods section (page 7): "The CCD detector measures the intensity in energy-

dependent instrument units. They are not calibrated to the x-ray photon count at the moment, and therefore are listed as arbitrary units in the figures.”

Reviewer #3. Color scales should be attached also to Figs. 4 and 5.

Our Response. Done.

Reviewer #3. Because the scattering angle was about 120 degrees, the magnification should be anisotropic in all the obtained CCD images. I hence recommend that the distorted direction should be explicitly mentioned. For example, two-dimensional length scale bars could be attached to Figs. 3, 4, and 5.

Our Response. This is a very good point. Our original text mentioned the need for x-ray beam footprint correction of the images, but did not give any explicit description. Indeed, the vertical length scale in the original detector images is different by 15% from the horizontal one. There are two ways (both good, we believe) to fix this: add an additional vertical length scale bar to the detector images, or stretch all the images vertically by 15% and therefore make the magnification uniform and the length scale bars identical. We chose the second approach, which we believe leads to better visual image comprehension. Specifically: all detector images in the paper are now stretched vertically by 15%. This is described explicitly in Methods, as well as in Fig 3 caption. Added text: Figure 3 caption “All the detector images in this paper are elongated vertically by the factor of 1.15 to compensate for the beam footprint size effect and produce uniform magnification, see the Methods section.” Methods section, page 7 “In this geometry, the direct image of the sample surface observed on the detector is compressed along the detector’s vertical direction by the factor of $\sin(\theta)=0.87$ due to the beam footprint size effect. All the detector images shown in this paper are elongated vertically by the factor of 1.15 to compensate for this compression. Thus, the images feature identical length scale bars for the vertical and horizontal directions (uniform magnification).”

As a final comment, we’d like to collect together the general statements of all the Reviewers regarding their recommendation to publish our work in Nature Communications, which we believe are all very positive. Reviewer #1 “I am supportive of the publication of this manuscript.”, Reviewer #2 “In my view, the paper reports important and convincing new results and is of interest for the readers of Nature Communications. In my opinion, the paper is publishable in the present form.”, and Reviewer #3 “I think that this finding is worth publication in Nature Communications.”

REVIEWERS' COMMENTS:

Reviewer #3 (Remarks to the Author):

I read the authors' response and revised manuscript. I am pleased to know that the authors have addressed all the comments in a satisfying way. The proposed imaging method of antiphase boundaries of antiferromagnetic order are quite interesting. Their discussion about the comparison with other imaging methods is also useful. Now I recommend that the paper should be published in Nature Communications.

We are delighted to see that Reviewer #3 now recommends our paper for publication, with no further modifications requested.

REVIEWERS' COMMENTS:

Reviewer #3 (Remarks to the Author):

I read the authors' response and revised manuscript. I am pleased to know that the authors have addressed all the comments in a satisfying way. The proposed imaging method of antiphase boundaries of antiferromagnetic order are quite interesting. Their discussion about the comparison with other imaging methods is also useful. Now I recommend that the paper should be published in Nature Communications.